# The lived experience of long COVID: A thematic analysis of an in-depth interview study

Zoe Sirotiak[1,2,3]*, Hailey J. Amro[2,3], Emily B. K. Thomas[3]

**1** Department of Kinesiology, Iowa State University, Ames, Iowa, United States of America, **2** Department of Psychology, Iowa State University, Ames, Iowa, United States of America, **3** Department of Psychological and Brain Sciences, University of Iowa, Iowa City, Iowa, United States of America

* zmsiro@iastate.edu

## Abstract

Long COVID is associated with significant physical and mental health burden, resulting in substantial quality of life limitations. The lived experience of individuals with long COVID is a vital consideration in evaluating the impact of the condition. Thirty-four adults with self-reported long COVID participated in a semi-structured in-depth interview study. An interview guide assessed physical and mental health symptoms, changes to plans, goals, and beliefs, and social impacts of long COVID. Participants were an average age of 51.6 years (SD = 17.0), and most identified as female (61.8%), white (97.1%), and not Hispanic or Latino/a/e (97.1%). Two coders read each interview while creating a codebook. The coders individually coded each interview transcript with themes emerging from the coded interviews. Each code reached an agreement level of at least 80%, with a Kappa (RK) score range of 0.90 to 0.98 in each interview, indicating adequate interrater reliability. Five themes emerged from the thematic analysis: decreased autonomy, decreased trust, changes in worldview, social impacts, and uncertainty. Individuals with long COVID reported heterogenous experiences, with often significant changes to daily functional abilities and outlook on life. Considering the unique lived experiences of individuals with long COVID will be important in developing a complete understanding of the condition and its implications.

## Introduction

Since the origins of the COVID-19 pandemic in 2019, heterogeneity in symptoms and severity has been observed in acute SARS-CoV-2 infection, with higher acuity in elderly populations and in populations with comorbid health conditions [1,2]. Severity ranges from asymptomatic to severe symptoms requiring hospitalization or resulting in fatalities [2]. Similarly, there is increasing evidence that the long-term impacts of SARS-CoV-2 are heterogeneous [3]. It has been estimated that between 6% and 39% of individuals experience prolonged symptoms after acute SARS-CoV-2

**Data availability statement:** The data supporting the findings of this study include code frequency tables, an interview guide, codebook, and de-identified illustrative quotes from participant interviews. These materials are publicly available on the Open Science Framework at https://doi.org/10.17605/OSF.IO/6ZW4H. Full interview transcripts are not shared due to ethical and confidentiality considerations as approved by the Iowa State University Institutional Review Board.

**Funding:** The author(s) received no specific funding for this work.

**Competing interests:** The authors have declared that no competing interests exist.

infection [4,5]. Suggested prevalence rates of long COVID have varied substantially, influenced by the novelty and multisystem impacts of the condition [4,5]. Past literature has varied regarding the criteria used to define long COVID, further contributing to the variation in prevalence noted [4,5]. Much of the current research considers the physical and mental health implications of long COVID or long-term sequelae of SARS-CoV-2 [6]. Common symptoms of long COVID include shortness of breath, fatigue, joint pain, chest pain, and sleep difficulties [6,7]. Moreover, SARS-CoV-2 infection is associated with increased risk for long-term consequences across several physical and psychological systems, including cutaneous, respiratory, cardiovascular, musculoskeletal, neurological, mental health, and renal domains [8,9]. Even those with mild severity acute infections have experienced long-term physical and mental health consequences [10].

There are nuanced biopsychosocial impacts of long COVID, including living with chronic physical and mental health concerns, slow rehabilitation and recovery, limited knowledge of long-term effects of SARS-CoV-2 infection, limited social support, limited access to healthcare services, and unsatisfactory experiences with healthcare providers [11]. These biopsychosocial impacts have been outlined in prior meta-analyses of qualitative study of long COVID experiences [11]. Other literature shows the heterogeneous nature of the long COVID experience, highlighting the continued need for qualitative research that gives voice to the specific experiences of those living with long COVID. Furthermore, prior research has emphasized the necessity of connecting individuals with long COVID to various resources and systems of support, based on identified barriers to healing [12,13]. These experiences and long-term impacts, alongside physical and mental health symptoms as well as the continued need for supportive resources, poses a need for further understanding of individual experiences of long COVID to more effectively treat and support those impacted by the condition. Such investigations will provide essential information about the mental, social, and physical health impacts of long COVID, and this information is necessary to develop and test interventions.

The aim of this qualitative study was to examine the lived experiences of those with self-reported long COVID, as prior research has shown the benefit of taking a phenomenological approach in understanding chronic illness [14]. Considering the lived experience, related to the subjective experience of an individual including psychological factors such as emotions, goals, and perceptions of an individual at a given point in their lives [15], had particular advantages for our study. In alignment with phenomenological research, our study hopes to gather information on patterns of experiences among people with self-reported long COVID, through highlighting individual experiences [16]. In addition, the study seeks to explore the biopsychosocial impacts of long COVID, gaining a deeper understanding of the functional, quality of life, and existential impacts of long COVID from the perspective of those directly impacted. This biopsychosocial framework was adopted in addition to the phenomenological approach, to highlight the systemic and compounding impacts of biological, social, and psychological experiences of those with self-reported long COVID [17]. The combination of these two approaches allows for a more holistic view of the long COVID experience.

## Materials and methods

### Ethics statement

The project was approved by the Iowa State University Institutional Review Board (IRB ID: 23–158). Formal written or verbal informed consent was not required by the Iowa State University Institutional Review Board (IRB) as this project was deemed exempt. Participants read through a consent information sheet and indicated consent by clicking forward on the screen. Participants were encouraged to reach out to the research team with questions or concerns at any point during the research process. Participants also gave verbal consent for recording their interview prior to beginning the interview. The survey was completed prior to the interview in all cases to allow participants to read information about the study and decide whether they would like to participate prior to engagement in the interview process. Participants were free to skip any questions that they did not wish to answer during both the survey and the interview.

### Participants and data collection

Semi-structured qualitative interviews were conducted over telephone with a sample of individuals with self-reported long COVID (*N* = 34). Participants were recruited from a prior online survey study probing the experiences of individuals with long COVID [18]. The parent study recruited participants through mass email and long COVID-specific social media sites. Individuals who completed the parent survey and self-reported long COVID had the option to offer name and contact information for future studies. These individuals were recruited via email for interviews, with other participants snowball sampled from prior participants. The recruitment period occurred from May 15, 2023, to August 1, 2023. The N = 34 was determined through the availability of funds and participants indicating interest in the interview, exceeding the number of interviews typically required to achieve saturation [19]. All eligible participants indicating interest were included.

Inclusion criteria included being ≥18 years old, being from the United States, English verbal communication abilities, and reliable internet and telephone service and devices. Prior to the interview, participants responded to a survey assessing sociodemographic characteristics such as age, gender, race, ethnicity, highest level of education, and income as well as several physical and mental health characteristics. An interview guide was prepared to probe for specific aspects of the lived experience, including physical and mental health symptoms and changes (*"Which symptoms have been most impactful?"; "How do you feel your mental health has changed over the course of having long COVID?"),* how plans, goals, and beliefs have changed since developing long COVID (*"How has your experience with long COVID impacted your plans and goals?"; "Do you feel like you are the same person as you were before your infection?"),* as well as the social impact of long COVID (*"What has been the reaction of others in your life such as bosses, coworkers, or acquaintances?").* The interview guide and additional information about the interview study are available on OSF [18].

Total interview durations ranged from 38 minutes to 3 hours and 12 minutes. Participants were an average age of 51.6 (SD = 17.0) years, with most identifying as female (61.8%), white (97.1%), and not Hispanic or Latino/a/e (97.1%). Most participants had achieved at least a bachelor's degree (79.4%), and nearly half of the participants (47.1%) reported earning greater than $100,000 per year.

### Qualitative analysis

Analyses involved two coders. One coder was involved in project conceptualization and data collection, whereas the other coder was involved in coding and analysis only. An inductive thematic analysis method was used to code the interview transcripts. The two coders individually read the complete interview transcripts in sets of five randomly selected interviews at a time. The coders collected potential codes and discussed between rounds of coding to begin establishing a codebook, combining and condensing codes noted individually. Although code saturation was assessed after every 5 interviews, subcode saturation was not met, with each set of 5 interviews providing at least one new subcode, perhaps reflecting the heterogenous nature of long COVID [19–21]. However, it has been argued that saturation may not be

required for impactful qualitative research [22]. Despite not reaching subcode-level saturation, all primary codes and final themes had emerged prior to the final set of interviews. After reading through all 34 interviews, the coders established a primary codebook and reviewed the interviews again to ensure that coding was complete [19–21].

Following establishment of the codebook, the coders individually coded each interview transcript. MAXQDA software [23] was utilized to code each document and compare across coders. Following independent coding, interrater reliability was assessed in MAXQDA. Codes with agreement of <80% with both unassigned and assigned codes counting as matches were reconsidered and discussed, with recoding resulting in agreement of at least 80%. Kappa (Kappa (RK)) [24] was utilized to assess agreement within each individual transcript across coders. The agreement by chance is 0.5, with a higher kappa score indicating higher levels of interrater reliability [24]. Following completion of coding, the coders jointly identified themes emerging from the transcripts. The range of agreement for individual codes ranged from 82.4% to 100%, with an average agreement score of 92.5%. The Kappa (RK) scores for individual interviews ranged from 0.90 – 0.98.

## Results

Five overarching themes were identified through discussion between the coders following completion of coding. These five themes are *decreased autonomy, decreased trust, changes in worldview or outlook on life, social impacts, and uncertainty.* Code frequencies, descriptions, and example quotes can be viewed in Supplementary Tables S1–S22 Tables and on OSF [18]. Examples of subthemes and codes driving theme emergence can be viewed in Fig 1.

### Decreased autonomy

Participants noted varying functional deficits both physically and cognitively as a result of long COVID. Many of these deficits impeded participants' ability to function within several domains, including interpersonal, work, school, and leisure functioning. For some, this included physical limitations and physical dependence on others. For others, this resulted in a change in future goals and plans. Many spoke about frustration and grief related to this reduced autonomy as exemplified below:

> "…there's definitely a lot of grief in that process… that independent person that I used to be, just isn't. I mean, I'm still very, I want to be independent, but I can't; I have to rely on others a lot more now. And so that was a big spot with grief."

> "…I feel like I have less like autonomy. I think [I have the same perception of how the] world works, but I think I have less autonomy now that I get to decide if that happens or not, because I can't…."

Participants spoke on the specifics of reduced functioning. They also noted a variety of physical impacts, including no impact, being limited in daily tasks, and being housebound or bedbound. This highlighted the diversity in experiences, as well as diversity in the degree of physical limitations and subsequent assistance needed from others:

> "I'm dependent on someone else to do my ADLs [activities of daily living] now."

> "I got to the point where I was fully bedbound. Sometimes someone would have to walk me to the bathroom. I could not brush my teeth and shower in the same day."

Many participants also noted that the condition impacted future goals or plans. Some noted slower or adjusted progress toward meeting goals, with little to no change in the goals themselves. Others described that they were unable to complete job tasks, live in the same place, or engage in the same activities. This further highlighted the diversity in experiences and variability in lasting impacts of participant symptoms and functional deficits due to long COVID:

## Themes and Selected Subthemes and Codes

### Decreased Autonomy
Physical limitations
Physical dependence
Change of future plans/goals
Bedbound/housebound

Change in career
Financial limitations
Reliance on others' perceptions
Few available treatments

New comorbidities
Daily routine disrupted
Adjusted activities of daily living
Increased time managing health

### Decreased Trust
Healthcare system
Medical providers
Employers and coworkers
Family and friends

Political system/government
Religious and spiritual beliefs
Change to alternative medicine
Health trajectory

Research funding
Intentions of others
Coping strategies
Resentment

### Changes in Worldview
Morals/Values
Religiosity
Role of medical system
Health trajectory

Focus more on self
Life purpose
Societal support
Views of others

Importance of support system
Perception of self
Outlook on life
Safety

### Social Impacts
Emotional support
Frustration
Misguided advice
Disbelief

Psychological attribution
Poor understanding of long COVID
Physical assistance
Better/worse social connections

Less social interaction
Support groups
Visibility of long COVID
Judgement of long COVID/precautions

### Uncertainty
Knowledge of long COVID
Treatment options
Personal cause of long COVID
Financial impact

Physical activity response
Symptom fluctuation
Impact of comorbid conditions
Employment status

Reaction of healthcare providers
Long-term impacts of long COVID
Realistic plans and goals
Disability status

**Fig 1. Results of the thematic analysis presented herein: An abbreviated summary of themes, subthemes, and codes.** *Note.* Themes are presented in the largest text, with subthemes and codes presented within each theme. Frequencies and example quotes can be found in Supplemental Materials S1–S22 Tables.

"I still do have potential for, you know, making a career I enjoy. It's just going to be different than what I envisioned… Personally that's a little harder because I've been trying to get, you know, healthier and in better shape and it's been very slow going with the struggles that I've been facing… That's frustrating for personal goals, but for career goals, I feel like I can still, I can make it work, you know?"

"I had to move houses. I had to move in with my parents."

Other participants referenced positive changes to goals and plans. This included an increased focus on spending time with loved ones and increased focus on health, in some cases by necessity. This also manifested as finding new hobbies and finding a new purpose in life due to experience with long COVID. This further highlights the diversity in perspectives experienced by participants, as some were compelled to recognize the positive impact of change within functional abilities:

"And even like just a little thing like hobbies that I'm doing now, like the Nintendo thing and like I've been crocheting and stuff that I really couldn't do before. So just enjoying even though I have limitations, enjoying what I can and [am] capable of and finding ways to utilize that for some new hobbies."

"I'm trying to do a lot more self-care now than I used to. I've found that it's become more necessary."

**Decreased trust**

Participant interviews highlighted a decrease in trust in several domains, including the healthcare system, larger societal systems such as the government, and within interpersonal relationships. Many participants linked this reduced trust to experiences with institutions while managing long COVID. Much of the mistrust in the healthcare system was attributed to participants' negative interactions with medical providers and the healthcare system. Many participants noted a perception that the healthcare system and medical providers were unable or unwilling to provide them with the support they needed:

"I've had to give up on doctors. Doctors don't believe me… the last doctor I saw, believed me... it was just like, well, I believe you, but good luck. I don't have anything to do with that, so I can't help you. And I don't even know how I would help you, even if I could… and I apologize."

"I've definitely found myself a lot more… compassionate to anyone who's not healthy and definitely a lot more angry and frustrated at the state of, you know, the way medicine works in this country and all the countries and how it's just not really helping anybody."

Others noted increased awareness of the barriers that exist systemically for disabled or chronically ill individuals. Some participants described personal recognition of long COVID as a disability.

"The world isn't really set up for disabled people… that's something that I was oblivious to having nobody really close to me need a wheelchair or anything like that. But it's very interesting to see from the other side how people treat you and how things just kind of aren't set up for you."

"It is a disability, you know, it's something that, that makes you incapable of doing what you did before. And, you know, I fortunately have the kind of job where, you know, I can rearrange things so I can keep working. I don't think it's going to be any problem for me to keep going another nine semesters to retirement. Um, but I know there are plenty of people who have different kinds of jobs who, um, who probably can't work anymore."

Participants reported decreased trust in governmental systems as a result of long COVID experiences. Many reported frustrations with perceived misinformation and mishandling of the COVID-19 pandemic by governmental organizations. Some participants noted contextual factors, such as the political climate and divisiveness within politics at the time of the COVID-19 pandemic, as contributing factors to reduced trust:

"I have sort of much less trust in the government or the authorities or doctors or, you know, people you're supposed to trust to help you out and take care of things. I don't view anybody as competent or good at their jobs anymore. And probably prior… I never really gave it much thought or question. You know, it seemed a doctor knew what they were doing."

"You know, I'm probably more suspicious of governments than I was before."

Participants also described increased mistrust in interpersonal relationships. Some participants described this experience as recognizing more superficiality in relationships. This also impacted the participants' worldview and views of society (further described in Social Impacts subsection below).

"I don't trust people that much anymore… I don't trust many."

"And I don't blame anyone, but you just learn that [things were just] surface level with some people."

## Changes in worldview

Participants reported several changes in perceptions of the world because of experiences with long COVID. Many noted changes to worldviews, changes in morals and values, and changes in religiosity throughout the trajectory of long COVID. These changes occurred at different time points and were prompted by several factors, each individual to the participant. Notably, many participants described changes in morals and values, as opposed to describing morals or values strengthening or weakening. It was also evident that these changes were related to changes in lifestyle and ability due to long COVID:

"I changed a lot of my morals. I had to switch my whole entire life around and kind of just look at a whole new perspective of life, I guess."

"Yeah, I think a lot of people… put a lot of value on productivity and what you can get done in a day. And that can directly or indirectly be related to a sense of self-worth. So that was something that I really had to like struggle through, especially at the beginning…"

Participants also reported changes in religiosity because of long COVID. Some participants noted a decrease in religiosity, described as resentment for religious belief systems. Other participants noted an increase in religiosity. Participants also shared that religion provided them with perspective, offering a coping strategy throughout difficulties with long COVID. However, most participants noted experiences with long COVID had little or no impact on religious beliefs and/or religiosity, and reported maintained religious views or lack thereof:

"No, I'm a spiritual person but not a religious person, and I don't go to church. I was raised going to church, and I just don't go, but I still have the same sort of spiritual relationship that I always had before and beliefs. Nothing really changed there through the course of it."

"I was an atheist, and I'm still an atheist."

Participants also described broad changes in worldview. Participants noted positive and negative perspective changes, which varied by participant and were defined through the participants' perception of the change. Some participants noted worldview changes in increased ability to enjoy things, improved optimism, increased desire to remain present, increased focus on themselves, and taking less for granted. Other participants noted increased pessimism and an increased notion that the world is unfair. However, there was also nuance in how individuals perceived these worldview changes:

"I think it changed my outlook on [self-care]. It made me realize that I wasn't going to be… immortal in a body that couldn't be broken."

"I have no idea what's going on in the world anymore. So it has shrunk to my world as myself and my immediate family as much as I can. [Previously] I was very, very active in things that were going on. Now, [I'm] not… So my worldview is myself."

## Social impacts

Participants had varied experiences with social support while managing long COVID. Some reported positive social interactions and support within interpersonal relationships. Others reported frustration with the social support offered by family,

friends, coworkers, and other social relationships. Many participants noted recommendations for improved social support including improved education, increased respect for functional limitations, increased sympathy, and willingness to offer assistance through long COVID. Notably, several of participants reported a significant decrease in social interaction, due to symptoms:

> "And so I've had to stop doing things or have chosen to stop doing certain things, especially a lot of after hours social stuff."

> "I mean, because there is days literally, there's days that I can go 24 hours without opening my mouth and speaking a word to another human being."

Participants also reported on several satisfactory social support interactions, including physical assistance, patience, and accommodations to shared activities. Participants noted satisfaction with others' willingness to educate themselves about long COVID, emotional support, and validation or belief in symptoms:

> "Um, you know, like I said, everybody has been pretty supportive. I haven't had to deal with anybody, but yeah, I think it helps that for the most part, friends and family are all highly educated. And so, you know, it's not like I have anybody in, anybody who's, I'm in a close relationship with who, who doesn't believe in this stuff."

> "Um, a lot of people have been very supporting, almost all actually. I did have some people, like say that I was faking it and stuff, but that was all years ago. So it doesn't really matter anymore, but almost everyone was very supportive, even though they didn't understand it."

Conversely, many participants noted unsatisfactory social support interactions. These interactions often had themes of disbelief in long COVID or the impact of long COVID symptoms. This manifested as others misunderstanding functioning limitations, providing misguided advice, attributing symptoms to psychological factors or other environmental factors, and placing blame on participants for long COVID symptoms. Other participants noted dissatisfaction with the pity they received from others. Participants expressed frustration from invalidation, and some noted decreased trust in their appraisal of reality:

> "And my stepmom insinuated that I needed like mental health medication and brought up a history of like family mental illness and I just thought like, you know, I don't know, like, am I losing it?"

> "I don't know that there is a single person in my life who believes that I have it. [...] And I feel as I get better or my symptoms get less severe, the less they believe I have it or I have ever had it."

These themes of satisfaction and dissatisfaction were also revealed in employment and academic social relationships, with participants noting appreciation for the availability of accommodations and flexibility within the workplace or in academic realms. However, similar unsatisfactory interactions, including lack of understanding of limitations and symptoms and lack of knowledge about long COVID, were also present. Some participants noted encouragement from employers to resign or the expectation to return to prior level of functioning:

> "Uh, with my, uh, with my bosses and coworkers, I think at first there was a lot of understanding when, you know, when I was fresh off of having the major infection. Uh, and then I think there's a lot of frustration as, as I wasn't being able to be as helpful. Um, cause as I said, my, my job performance physically dropped off. Um, and they, I know my boss, uh, in my former position was very understanding about it, but you know, there was just, there were certain expectations of things I needed to be able to do. Okay. So, so understanding, but frustrated."

"Yeah, they were not very understanding. Um, and they tried to, like, I don't know, coerce or like threaten me to come back to work. And I was also told that I needed to like resign or quit."

Participants reported varied interactions with other support systems, including social media platforms and long COVID support groups:

"I learned that you can't join still COVIDing [still taking precautions to avoid COVID-19] groups with long COVID because most of those people, and great for them, have never had COVID, so they don't understand. They don't understand, and also, there's a stigma. There's a stigma to the people who have never had COVID. There's a stigma on those of us who have gotten it, even though I think they're stating now it's like a good 98% of the population, which is weird. It's weird. The stigma put on COVID is weird. It is so weird. I think it was always weird."

"I have a very good Twitter long COVID group. I have one, a care group on Facebook, and additionally at [school], we have a group of physicians, nurses, and mid-level providers who all got long COVID, and we communicate."

Additionally, participants also noted themes of judgment and visibility concerns in experiences with others. Participants noted judgment from others around the general topic of long COVID or regarding safety precautions they adapted or maintained. Speaking to the variability in long COVID symptomatology, many participants noted difficulty with the (in)visibility of long COVID symptoms. Some reported wishing long COVID symptoms were more visible, and some wished symptoms were less visible. Participants noted frustration with others' (e.g., friends, family, employers, physician) dismissal of milder symptom profiles:

"I can't do the normal things. And then society thinks I should be able to. Like I look normal."

"I think like I say, the big thing is that people with the milder symptoms, I think can get neglected and looked over. The people that have more like the breathing issues and things like that, it might be that people are more concerned with them and work towards getting them to a better place. Well, maybe, like I said, the dizziness and the vertigo or anything else, those milder effects, it's like, yeah, you know, live with it."

Secondary to these interactions, several participants noted changes in perceptions of others, including society more broadly. This was described as lower tolerance for society and others, as well as decreased hope related to humanity:

"I don't know if our society cares about other people anymore really because it doesn't seem like anybody cares about anybody else period, whether it's you're sick or you're not…"

"People are mean, like my worldview, I guess that's what you're speaking of. People are awful. They're awful. They're nothing like, I mean, my hope for humanity has left since I've had long COVID. I stopped recycling because I don't want people to live. I want the world to live, I want the world to live, but I don't want people to live. I mean, it sounds crazy, but I don't. I don't want humanity to survive. Like I don't."

### Uncertainty

The varied nature of long COVID experiences contributed to uncertainty and distress throughout managing long COVID. Participants noted variability in prior knowledge of long COVID, the trajectory to long-term symptomatology, the trajectory of the illness, as well as a range of symptoms, treatments, and interactions with the healthcare system. Some participants described no prior knowledge of long COVID, and others reported limited knowledge or incorrect perceptions of the duration, severity, or symptom types of long COVID:

"I perceived it to be people that had the more serious forms of COVID than what I had."

"If you didn't go to the hospital, I didn't think it was going to be an issue, I guess."

As symptoms persisted, the progression into long-term symptoms differed across participants. For some, acute SARS-CoV-2 infections persisted into long-term symptoms, whereas others improved after the initial infection and saw a decline in functioning over time or after subsequent SARS-CoV-2 infections. Many participants reported on the unpredictability of the trajectory to long-term illness:

"I didn't know what was going on. I had no idea, but it was three weeks after infection. And I had no idea that all these symptoms could come up three weeks after infection and be caused by my COVID infection. Like I had no clue. So at the time, no, I didn't think it was long COVID… We had no idea what was going on..."

"So it was like, I must've been either, you know, I had it and it was before the testing or something, but it took a few months after like getting sick before I felt the long COVID symptoms start."

Throughout the duration of long COVID, participants also described uncertainty in the trajectory of long-term symptoms, sometimes causing significant distress. However, other participants noted hope for improvement in the future:

"But that's also part of the grief is figuring out whether to accept like, is this a long-term thing, is this the rest of my life thing, or is this something that I can say like has an end and I'll be able to be a normal person at the end of it?"

"I was so sure I was going to get better. I still think I will get better-ish. I just don't know that I can get back to the level of functioning that I was before, both physically and mentally, because I know that my cognitive abilities are not what they used to be. I don't know if that can be improved or not, or maybe I'm just stuck with this."

Participants also described a variety of symptoms and symptom fluctuation over time. Participants noted difficulties with predicting symptom severity day to day, as well as difficulty identifying activities that trigger or relieve symptom impacts. This seemingly further contributed to the overall sense of uncertainty related to experiences with long COVID:

"Honestly, it's kind of more frustrating because it's like, I start off the day, it's like, okay, is it going to be fine all day? Is it going to be fine until some point? Is it going to start off bad and then I get fine later on, but it's pretty unpredictable."

"Yeah, it's hard to say, it's really hard to say, because there's, I've tried so many ways at this point, it's been three years, I've tried so many ways to stabilize things and make things feel better and whatever, and they're really, it's really hard to put a finger on any one thing that helps and that maintains, you know, a good day."

Furthermore, participants reported a diverse range of prescribed treatments for long COVID with varying degrees of success. A common trend, however, was being referred to several providers and specialists, often with conflicting or unsuccessful treatment recommendations. Many participants reported on medical providers' lack of knowledge regarding long COVID due to the novelty of the condition, as well as lack of available research on long COVID. This contributed to feelings of frustration and uncertainty among participants:

"But they weren't finding anything because it was like not a root of the problem kind of thing. So it was a lot of just after that, a lot of going to different specialists and [getting results] of normal."

"And I wish, I mean, if there are doctors out there who are doing more specific things or might have more breakthroughs, I wish they would research them and refer more. But a lot of them have just been like dead ends, but I know they don't have to be."

## Discussion

This qualitative study illustrated the experiences of a sample of individuals with long COVID and the biopsychosocial impacts of the illness among this sample. Our findings build upon the existing literature, which highlights the long-term and impactful nature of long COVID symptoms [11]. Our findings indicate that long COVID may be associated with existential changes in worldview, as well as increased mistrust in social systems, including healthcare, employment, education, and interpersonal relationships. However, uncertainty in the perspectives of those with long COVID was also common in relation to treatment, recovery, and long-term outcomes, and this uncertainty adversely impacted mental health. Like other chronic illnesses, long COVID appears to have significant and debilitating impacts on physical abilities, daily functioning, and engagement in activities [25]. Moreover, long COVID may prompt feelings of shame, fear, and uncertainty, all of which may impact an individual's quality of life [26,27]. Identity loss and reconstruction as well as biographical disruption have been frequently noted in the context of chronic illness [28], and individuals with long COVID may struggle with understanding their own biography and place in the world after substantial disruptions to their body and daily life. These findings indicate that individuals with long COVID may not only struggle with immediate mental health symptoms but more existential, long-lasting doubts about their role and future in their new experience.

As individuals with chronic illness develop new insights and have access to new resources, prior research indicates that they can experience positive changes to perspectives on life [26]. Some of these resources include experiencing social support, developing new ways to do things, having certain personality characteristics such as optimism and patience, having confidence in the future, cultivating illness acceptance, and having increased value of life [26,28]. Our findings were similar; in that many participants who experienced positive support from others, adaptations to account for functional limitations, and acceptance of the illness noted positive changes in worldview and life perspectives. Other participants noted more negative changes to worldviews and perspectives. From the lens of these adaptability resources, long COVID management may be complicated by systemic and environmental factors. For example, many participants in this study reported a lack of support from social connections, family members, and others due to the unclear nature of the condition and the lack of research and knowledge available regarding long COVID. The lack of available information regarding symptoms and treatments of long COVID may perpetuate uncertainty for those impacted, effectively limiting feelings of hope. Furthermore, as suggested by the literature, stigma surrounding long COVID may further negatively impact mental health [29], indicating that improved resources may be necessary for more effective adjustment among individuals living with long COVID.

Additionally, many participants reported a sense of mistrust in several social systems. Medical stigma has been associated with feelings of invalidation, distrust, isolation, and rejection, limiting treatment and prevention effectiveness within healthcare systems [30]. It has also been noted that the social stigma surrounding long COVID correlates with poorer mental health and mental health-related quality of life [31]. Stigma, enabling discrimination and reducing social acceptance, has been associated with several negative impacts including impaired mental health, social relationships, and access to opportunities [32,33]. The Health Stigma and Discrimination Framework, developed to apply to a range of medical conditions including leprosy, cancer, and obesity, describes drivers of stigma (e.g., beliefs that individuals with mental health issues are dangerous), facilitators of stigma (e.g., perception that individuals with mental health issues are incompetent), as well as intersecting stigmas (such as race, gender identity, sexual orientation) [34]. These contributors to stigma impact manifestations among affected individuals (e.g., discrimination, negative public attitudes) as well as outcomes (e.g., delays in accessing care, employment restrictions, or health insurance coverage) [34].

Negative experiences with the healthcare system have been reported among individuals with long COVID. These negative experiences have been attributed to several barriers to effective care, including limited long COVID knowledge, limited access to holistic care, and limited symptom and lived experience validation [12]. This prior work is corroborated by our findings, in which many participants experienced significant stigma from not only healthcare systems, but employers, academic institutions, family members, friends, and others. This stigma sometimes subsequently limited participant

willingness to seek care, and perpetuated feelings of isolation, invalidation, and distrust within these systems. These consequences and mistrust may contribute to reduced effectiveness in treating and preventing long COVID in the future.

Our findings may also indicate that there is a level of uncertainty related to long COVID outcomes among individuals living with the condition, with this uncertainty exacerbated by the lack of research and knowledge available about long COVID, as well as the uncertainty surrounding available treatments. Participants reported conflicting treatment recommendations from healthcare providers, as well as contradictory information regarding long COVID symptoms and treatment. Past literature has shown that illness uncertainty is related to poor adjustment, reduced tolerance to painful stimuli, maladaptive coping, higher psychological distress, and reduced quality of life [35]. The biopsychosocial impacts of living with long COVID may be related to significant uncertainty surrounding long COVID and its trajectory. The substantial uncertainty noted by our participants was similar to the consideration of medically unexplained symptoms or medical conditions without known organic cause, which are frequently associated with long COVID [36,37]. Individuals with medically unexplained syndromes and medical providers treating individuals with these conditions have frequently noted uncertainty regarding several aspects of these conditions, including causes and ideal treatment [38,39]. Individuals with long COVID note similar substantial uncertainty, affecting perceptions of their condition and expected future.

The substantial and disabling nature of long COVID symptoms was frequently noted throughout the interviews. In addition to the individual impacts of symptoms, participants often noted a change in their autonomy and function, sometimes noting that they now consider themselves to have a disability. Disability can be a social identity, indicating potential implications beyond the immediate impact of physical and mental health symptoms experienced related to long COVID. Other studies in the context of long COVID have emphasized the expansive nature of disability that can affect individuals, with the impact of physical symptoms compounded by barriers to social inclusion, cognitive symptoms, challenges in completing activities of daily living, as well as mental and emotional health problems [40]. Others with long COVID have noted the social implications of long COVID as an invisible disability, noting that others often are not accepting and understanding [41], influencing social relationships and the lived experience. Long COVID has been noted to substantially affect an individual's sense of self, often with notable identity loss [41], indicating that long COVID can impact much beyond the physical symptoms. Although some individuals with long COVID may note disability, it is also important to note that others may not identify as an individual with a disability.

As shown by our results and the long COVID literature, long COVID impacts biological, psychological, and social experiences of those impacted. These impacts vary in acuity and persist in both the short- and long-term. The variability in symptom profile and symptom impact demonstrates the need for diverse and individualized treatment and support options for individuals with long COVID. However, as conveyed herein, access to treatment and support may be limited by the novelty and complexity of this condition. The diverse nature of participant experiences, as well as variability in symptom profile, medical treatment received, and quality of life implications introduces a potential challenge in asserting a generalizable treatment recommendation for those with long COVID. However, this variability and diversity highlight the need for individualized care dependent on an individual's experience and symptom profile. Additionally, given this diversity in experience, it will be important to continue utilizing illness narratives from those experiencing long COVID, as they have been influential in the development of knowledge and advocacy for those managing long COVID illness [27]. These findings also emphasize inconsistencies in diagnostic procedures, which suggests a need for streamlined processes to diagnose long COVID. The perceived lack of education and understanding from medical professionals also stresses the need for increased dissemination of long COVID research among healthcare providers and the broader public. Continued awareness of SARS-CoV-2 infection will also be important for future monitoring and diagnosis of long COVID.

In this study, participants opted into the interview process voluntarily. This may introduce self-selection bias, such that individuals with more severe long COVID experiences or stronger opinions regarding long COVID may have been more likely to participate. However, individuals with very severe symptoms may not have been able to participate in the study and may not have their experiences represented. Participants were not asked about their motivation to participate in

the interviews, though this motivation may influence participants' likelihood of taking part in the study. Participants were recruited from a prior survey study that involved recruitment through social media and mass email, and individuals without access to these resources may not have been included. In addition, individuals with more social engagement with their illness experience may be particularly likely to be recruited from social media, biasing the participating sample. We utilized snowball sampling, and it is possible that participants may be more homogenous than the larger long COVID population given preexisting similarities between participants. Additionally, we did not require that participants provide medical records to verify diagnosis, which may have introduced inconsistency in the reported findings. It is possible that individuals reporting long COVID may be experiencing symptoms due to another medical condition or process. However, there is not established diagnostic criteria for long COVID and reports of long COVID often depend on patient self-report of symptoms [42]. Furthermore, the small sample size limits generalizability of the findings. However, the use of qualitative interviews allowed the authors to collect detailed accounts from participants regarding lived experiences with long COVID. The homogenous participant demographic characteristics further limit the generalizability of the findings. To participate in the study, participants had to have access to the internet and a telephone, and individuals without access to these resources would have been excluded from the study, limiting generalizability.

Our participants predominantly identified as female, white and not Hispanic or Latino/a/e, limiting interpretation to populations beyond this sample. Other work centered on long COVID has argued for the importance of promoting diversity and inclusion of diverse sociodemographic characteristics and experiences, with success incorporating community- and patient-engaged models [40,43]. Further work investigating the lived experience of long COVID from the experience of a diverse set of individuals with long COVID may provide additional insight into the variability of the impacts of long COVID. Sociodemographic characteristics such as race and ethnicity have been associated with stigma and health disparities [44], indicating the importance of considering how these factors influence the long COVID illness experience. Non-Hispanic Black and Hispanic adults were noted to have significantly higher odds of medical mistrust compared to non-Hispanic white individuals in a survey completed prior to the COVID-19 pandemic [45], emphasizing that sociodemographic characteristics and lived experience related to other identities must be considered when evaluating the experiences of individuals with long COVID. Non-white individuals and individuals without college degrees have been suggested to be more likely report long COVID, as well as activity limitations due to long COVID [46], emphasizing the potential limitations of applying our results to the wider long COVID population, given our heterogenous sample consisting mostly of white individuals with higher education. Individuals with Black and Hispanic identities have been suggested to have significantly greater odds of developing long COVID compared to white and non-Hispanic individuals, respectively [47]. Symptoms of long COVID have been suggested to vary significantly based on sociodemographic group, further emphasizing the potential limitations presented by our largely homogenous sample. The limited sociodemographic diversity within a substantial proportion of long COVID literature has been critiqued, with some noting that incorporating a more diverse set of characteristics will be important in developing a more complete picture of long COVID [48]. It will be important for future research to include a demographically diverse and representative sample in order to provide comprehensive insight on the needs within long COVID support services.

## Conclusion

Long COVID has been associated with significant symptom burden, with notable limitations in quality of life. This study was conducted following in-depth qualitative interviews with individuals living with long COVID. Five themes emerged from analysis of this wide-ranging interview, including decreased autonomy, decreased trust, changes in worldview or outlook on life, social impacts, and uncertainty. This sample of individuals with long COVID reported heterogenous experiences, emphasizing the importance of considering the individual experience in evaluating the impact of long COVID. Future research may involve longitudinal qualitative studies to further investigate changes in key domains over time among individuals affected.

## Supporting information

**S1 Table. Acute infection codes and frequencies.**
(DOCX)

**S2 Table. Comorbid conditions codes and frequencies.**
(DOCX)

**S3 Table. COVID susceptibility codes and frequencies.**
(DOCX)

**S4 Table. Daily functioning codes and frequencies.**
(DOCX)

**S5 Table. Extracurricular functioning codes and frequencies.**
(DOCX)

**S6 Table. Health before COVID codes and frequencies.**
(DOCX)

**S7 Table. Healthcare interactions codes and frequencies.**
(DOCX)

**S8 Table. Long COVID providers codes and frequencies.**
(DOCX)

**S9 Table. Long COVID theories codes and frequencies.**
(DOCX)

**S10 Table. Long COVID trajectory codes and frequencies.**
(DOCX)

**S11 Table. Mental health codes and frequencies.**
(DOCX)

**S12 Table. Perceived social support codes and frequencies.**
(DOCX)

**S13 Table. Perception of self codes and frequencies.**
(DOCX)

**S14 Table. Perspective changes codes and frequencies.**
(DOCX)

**S15 Table. Physical activity codes and frequencies.**
(DOCX)

**S16 Table. Physical health codes and frequencies.**
(DOCX)

**S17 Table. Plans and goals codes and frequencies.**
(DOCX)

**S18 Table. Prior knowledge codes and frequencies.**
(DOCX)

**S19 Table. Prior to long COVID codes and frequencies.**
(DOCX)

**S20 Table. Religious and spiritual ideology codes and frequencies.**
(DOCX)

**S21 Table. Symptom fluctuation codes and frequencies.**
(DOCX)

**S22 Table. Treatment for long COVID codes and frequencies.**
(DOCX)

## Acknowledgments

We are sincerely grateful to the participants whose insights and lived experiences with long COVID made this study possible.

## Author contributions

**Conceptualization:** Zoe Sirotiak.

**Data curation:** Zoe Sirotiak.

**Formal analysis:** Zoe Sirotiak, Hailey J. Amro.

**Investigation:** Zoe Sirotiak.

**Methodology:** Emily B. K. Thomas.

**Project administration:** Zoe Sirotiak.

**Writing – original draft:** Zoe Sirotiak, Hailey J. Amro.

**Writing – review & editing:** Zoe Sirotiak, Hailey J. Amro, Emily B. K. Thomas.

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
