## [Decision Letter · Decision Letter 0]

28 Aug 2025

PMEN-D-25-00287

The lived experience of long COVID: A thematic analysis of an in-depth interview study

PLOS Mental Health

Dear Dr. Sirotiak,

Thank you for submitting your manuscript to PLOS Mental Health. After careful consideration, we feel that it has merit but does not fully meet PLOS Mental Health’s publication criteria as it currently stands. Therefore, we invite you to submit a revised version of the manuscript that addresses the points raised during the review process.

please see the reviewers recommendation below for your action

We look forward to receiving your revised manuscript.

Kind regards,

David Onchonga, Ph.D.

Academic Editor

PLOS Mental Health

Journal Requirements:

1. Please send a completed 'Competing Interests' statement, including any COIs declared by your co-authors. If you have no competing interests to declare, please state "The authors have declared that no competing interests exist". Otherwise please declare all competing interests beginning with the statement "I have read the journal's policy and the authors of this manuscript have the following competing interests:"

3. Please insert an Ethics Statement at the beginning of your Methods section, under a subheading 'Ethics Statement'. It must include:

1) The name(s) of the Institutional Review Board(s) or Ethics Committee(s)

2) The approval number(s), or a statement that approval was granted by the named board(s)

3) (for human participants/donors) - A statement that formal consent was obtained (must state whether verbal/written) OR the reason consent was not obtained (e.g. anonymity). NOTE: If child participants, the statement must declare that formal consent was obtained from the parent/guardian.

4. Please provide separate figure files in .tif or .eps format.

https://journals.plos.org/mentalhealth/s/figures

https://journals.plos.org/mentalhealth/s/figures#loc-file-requirements

5. We have noticed that you have uploaded Supporting Information files, but you have not included a list of legends. Please add a full list of legends for your Supporting Information files after the references list.

Reviewers' comments:

Reviewer's Responses to Questions

**Comments to the Author**

1. Does this manuscript meet PLOS Mental Health’s publication criteria?

Reviewer #1: Yes

Reviewer #2: Yes

2. Has the statistical analysis been performed appropriately and rigorously?

Reviewer #1: N/A

Reviewer #2: Yes

3. Have the authors made all data underlying the findings in their manuscript fully available (please refer to the Data Availability Statement at the start of the manuscript PDF file)?

Reviewer #1: Yes

Reviewer #2: Yes

4. Is the manuscript presented in an intelligible fashion and written in standard English?

Reviewer #1: Yes

Reviewer #2: Yes

Reviewer #1: I greatly appreciate the topic as well as your choice of a qualitative approach, which constitutes an effective methodology for gaining a deeper understanding of the subjective experiences of long COVID patients.

However, there may be concerns regarding significant biases related to the sample. The experiences of individuals who are able to communicate and who have access to telephones and the internet are likely to differ substantially from those of individuals who are unable to communicate or who lack the necessary access to these technologies. Other characteristics of the sample—particularly gender and ethnicity—might also indicate a systematic bias. An important question is why participant self-selection occurred in this manner and how the sample differs from the broader population of patients with long COVID.

The severity of symptoms may also be an issue. It is possible that individuals with more severe long COVID symptoms or stronger opinions about the condition may have been more likely to participate. Conversely, individuals with very severe symptoms may not have had the capacity to take part in any research.

On the other hand, I do not consider the small sample size or the limited generalizability of the findings to be problematic, given that this is a qualitative study.

Considering the exploratory nature of the study, I would have expected the interviews to be less structured. Additionally, the use of percentages in the text is somewhat confusing.

More importantly, long COVID is an umbrella term encompassing various symptoms. The lived experience of each individual is inevitably shaped by their specific symptoms or combinations of symptoms. A stronger focus on the impact of particular symptoms on patients’ lives could provide a much clearer understanding of their experiences.

Good luck with your further research!

Reviewer #2: Introduction

1. While the introduction establishes the need for research into lived experience, it does not adequately review prior qualitative studies (if any) on long COVID or comparable post-viral conditions (e.g., ME/CFS). Positioning this study within a qualitative research landscape would enhance its originality claim.

2. The prevalence statistics provided are quite broad and could be problematized—what accounts for this variation (different populations, definitions of long COVID, time frames)? A brief clarification would improve scientific rigor.

3. The introduction states the aim to study "lived experiences," but does not define what is meant by this term or how it is conceptually grounded (e.g., phenomenology, medical sociology, illness narratives). Clarifying this would strengthen the theoretical foundation.

4. The introduction leans heavily on the biopsychosocial framing but does not make explicit why this model is chosen over others (e.g., disability studies frameworks, social determinants of health perspectives). Making this explicit could broaden interpretive richness.

Methods

1. The reliance on self-reported long COVID may reduce clinical validity, as no medical confirmation was sought.

2. This limitation is common in qualitative health studies but should be explicitly problematized.

3. Recruiting from social media and a prior survey may overrepresent individuals with more severe or socially engaged illness experiences.

4. Snowball sampling further risks homogeneity in participant perspectives.

5. The methods do not mention efforts to ensure diversity across demographics (e.g., race, socioeconomic status), which is important in understanding the varied impact of long COVID.

6. The authors note that code saturation was not reached, but themes did emerge.

7. While they cite literature suggesting saturation is not essential, a clearer justification of how they ensured analytical adequacy despite new codes emerging until the end would strengthen credibility.

8. Although exempt status was obtained, relying solely on participants clicking forward on a screen could be seen as minimal consent.

9. A stronger case could have been made for verbal consent at the beginning of interviews.

10. The methods do not describe what demographic data were collected (age, gender, race/ethnicity, socioeconomic background).

11. Since the study explores lived experience, demographic and contextual diversity are highly relevant for interpreting findings.

Results

1. The sample is overwhelmingly white (97.1%), highly educated (79.4% with bachelor’s or higher), and affluent (47.1% earning >$100k).

2. This homogeneity limits transferability of findings, as the lived experience of long COVID may differ significantly across racial, socioeconomic, and cultural contexts.

3. The frequent use of percentages (e.g., “23.5–26.5%,” “61.8–79.4%,” etc.) risks reducing qualitative nuance into quasi-quantitative reporting.

4. While this conveys prevalence, it can distract from the depth of narrative insight.

5. A more interpretive synthesis—prioritizing meaning over frequency—would align more strongly with thematic analysis principles.

6. Some themes (e.g., Decreased Autonomy) are richly developed with layered subthemes, while others (e.g., Changes in Worldview) are less elaborated in the provided excerpt.

7. This imbalance risks privileging some experiences over others.

8. The results focus on symptoms, autonomy, and trust but do not address how gender, race, class, or disability identity shape lived experience.

9. Given the sample’s demographic skew, the absence of reflection on this limitation is notable.

10. participants describe long COVID as a disability, yet the analysis does not explore the social or political implications of this identity shift.

11. This could have been developed more fully, perhaps linking to existing disability studies literature.

Discussion

1. The discussion occasionally frames the study’s findings as broadly representative of “individuals with long COVID,” despite acknowledging severe demographic skew and sample limitations later.

2. While the authors note demographic homogeneity, they do not explore how gender, race, or socioeconomic background might shape experiences of mistrust, stigma, or adaptation.

3. Given the sample characteristics, this omission leaves an important analytical gap.

4. The discussion references concepts like stigma, uncertainty, and existential change, but these are treated descriptively rather than fully theorized.

5. Engagement with frameworks such as illness narratives, disability studies, or medical sociology could have added richer interpretive layers.

**Do you want your identity to be public for this peer review?** For information about this choice, including consent withdrawal, please see our Privacy Policy

Reviewer #1: No

Reviewer #2: No

---

## [Decision Letter · Decision Letter 1]

22 Dec 2025

PMEN-D-25-00287R1

The lived experience of long COVID: A thematic analysis of an in-depth interview study

PLOS Mental Health

Dear Dr. Sirotiak,

Thank you for submitting your manuscript to PLOS Mental Health. After careful consideration, we feel that it has merit but does not fully meet PLOS Mental Health’s publication criteria as it currently stands. Therefore, we invite you to submit a revised version of the manuscript that addresses the points raised during the review process.

The manuscript has been evaluated by one reviewers, and their comments are available below.

The reviewer has two minor requests. Could you please carefully revise the manuscript to address all comments raised?

We look forward to receiving your revised manuscript.

Kind regards,

Steve Zimmerman, PhD

PLOS Staff Editor

Journal Requirements:

Reviewers' comments:

Reviewer's Responses to Questions

**Comments to the Author**

Reviewer #1: All comments have been addressed

publication criteria?

Reviewer #1: Yes

3. Has the statistical analysis been performed appropriately and rigorously?

Reviewer #1: Yes

4. Have the authors made all data underlying the findings in their manuscript fully available (please refer to the Data Availability Statement at the start of the manuscript PDF file)?

Reviewer #1: Yes

5. Is the manuscript presented in an intelligible fashion and written in standard English?

Reviewer #1: Yes

Reviewer #1: ¨Again, your theme as well as the qualitative approach is very interesting and provides new useful information.

I would appreciate even more detailed description of potential bias caused by relative ethnic cultural and socio-economic homogeneity of the sample.

Since it describes the methodology rather than results, the introductive part of chapter Results should be a part of Materials and Methods.

Good luck with yout future research!

**Do you want your identity to be public for this peer review?** For information about this choice, including consent withdrawal, please see our Privacy Policy

Reviewer #1: No

---

## [Editor Report · Decision Letter 2]

18 Jan 2026

The lived experience of long COVID: A thematic analysis of an in-depth interview study

PMEN-D-25-00287R2

Dear Dr. Sirotiak,

We are pleased to inform you that your manuscript 'The lived experience of long COVID: A thematic analysis of an in-depth interview study' has been provisionally accepted for publication in PLOS Mental Health.

Best regards,

Karli Montague-Cardoso

Staff Editor

PLOS Mental Health